# Exosomal Components and Modulators in Colorectal Cancer: Novel Diagnosis and Prognosis Biomarkers

**DOI:** 10.3390/biomedicines9080931

**Published:** 2021-07-31

**Authors:** Yu-Chan Chang, Ming-Hsien Chan, Chien-Hsiu Li, Chih-Yeu Fang, Michael Hsiao, Chi-Long Chen

**Affiliations:** 1Department of Biomedical Imaging and Radiological Science, National Yang-Ming University, Taipei 112, Taiwan; jameskobe0@gmail.com; 2Department of Biomedical Imaging and Radiological Science, National Yang Ming Chiao Tung University, Taipei 112, Taiwan; 3Genomics Research Center, Academia Sinica, Taipei 115, Taiwan; ahsien0718@gmail.com (M.-H.C.); dicknivek@icloud.com (C.-H.L.); 4National Institute of Infectious Diseases and Vaccinology, National Health Research Institutes, Miaoli 350, Taiwan; phildts@gmail.com; 5Department of Biochemistry, Kaohsiung Medical University, Kaohsiung 807, Taiwan; 6Department of Pathology, School of Medicine, College of Medicine, Taipei Medical University, Taipei 110, Taiwan; 7Department of Pathology, Taipei Medical University Hospital, Taipei 110, Taiwan

**Keywords:** extracellular vesicles, applications, regulation, biomarkers, colorectal cancer

## Abstract

The relatively high incidence and mortality rates for colorectal carcinoma (CRC) make it a formidable malignant tumor. Comprehensive strategies have been applied to predict patient survival and diagnosis. Various clinical regimens have also been developed to improve the therapeutic outcome. Extracellular vesicles (EVs) are recently proposed cellular structures that can be produced by natural or artificial methods and have been extensively studied. In addition to their innate functions, EVs can be manipulated to be drug carriers and exert many biological functions. The composition of EVs, their intravesicular components, and the surrounding tumor microenvironment are closely related to the development of colorectal cancer. Determining the expression profiles of exocytosis samples and using them as indicators for selecting effective combination therapy is an indispensable direction for EV study and should be regarded as a novel prediction platform in addition to cancer stage, prognosis, and other clinical assessments. In this review, we summarize the function, regulation, and application of EVs in the colon cancer research field. We provide an update on and discuss potential values for clinical applications of EVs. Moreover, we illustrate the specific markers, mediators, and genetic alterations of EVs in colorectal carcinogenesis. Furthermore, we outline the vital markers present in the EVs and discuss their plausible uses in colon cancer patient therapy in combination with the currently used clinical strategies. The development and application of these EVs will significantly improve the accuracy of diagnosis, lead to more precise prognoses, and may lead to the improved treatment of colorectal cancer.

## 1. Background of CRC

The high incidence of CRC makes it one of the malignances with the highest mortalities in the world [1]. In recent decades, many Asian countries have reported an increase in the incidence of CRC [2]. The low overall survival rate of patients diagnosed with advanced stages still cannot be improved despite recent advances in CRC screening and treatment. The morbidity and mortality of CRC after initial treatment are mainly caused by tumor recurrence and distant metastasis. Notwithstanding anti-vascular endothelial growth factor (VEGF) [3,4] and anti-epidermal growth factor receptor (EGFR) therapies [5,6], two targeted treatments currently available for CRC, relatively few means of improving survival have been reported. There is accordingly an urgent need to clarify the relevant mechanisms of tumor progression and find novel targets for their application in prognosis or treatment evaluation.

The primary lesions of colorectal cancer include bowel cancer, colon cancer, and rectal cancer. Most patients have symptoms such as abdominal pain, anemia, and bleeding. The development and derivation of aberrant polyps can eventually become CRC [7]. In addition, colonic epithelial cells are also prone to epithelial–mesenchymal transition (EMT). The EMT process is accompanied by drug resistance, cell morphology changes, metastasis, and the production/release of extracellular vesicles (EVs). There are many strategies to treat colorectal cancer (e.g., surgery, chemotherapy, radiotherapy, targeted therapy, and immunotherapy) and there are also complex combination treatments, such as folinic acid, fluorouracil, oxaliplatin (FOLFOX) and folinic acid, 5-fluorouracil (5-FU), and irinotecan (FOLFIRI) regimens [8,9,10]. Even with these approaches, colorectal cancer has not been significantly addressed. The inferior accuracy of early diagnosis and high metastatic instinct of CRC lead to a dismal survival rate among advanced patients.

In recent years, colorectal immunotherapy has become an additional standard regimen for patients with advanced diseases when the abovementioned chemotherapy/radiotherapy reaches its limit. Previous studies have reported the occurrence of mismatch repair defects (dMMR) in CRCs that resulted in hypermethylation in the promoter region of MLH1 gene [11]. This event led to an increase in high tumor mutational burden (TMB) as well as altered microsatellite sequences, leaving these tumors in a state of high microsatellite instability status (MSI-H) [12]. Therefore, TMB and MSI status have been considered as items to evaluate and select the appropriate population for immunotherapy [13,14,15]. More importantly, scientists are exploring how EVs derived from immune cells after colorectal immunotherapy can affect the immune response and promote immune system reprogramming. All in all, EVs contribute and connect the tumor microenvironment (TME) and local immune responses and play important roles in the development of CRC.

## 2. General Definition, Classification, and Application of EVs

Extracellular vesicles (EVs) are composed of lipid bilayers, which are mainly formed by cell secretion via exocytosis. These EVs can be classified into three subtypes, namely macrovesicles, exosomes, and apoptotic bodies, based on their release pathways, size, and content [16]. In this review, we mainly focus on the role of exosomes and cellular exocytosis in colorectal cancer. Exosomes carry mRNA, microRNA (miRNA), long non-coding RNA (lncRNA), protein, DNA fragments, and debris and transport them to the extracellular space [17,18,19]. Therefore, various studies have revealed significant exosome production in the TME, and the contents of these EVs are diverse. Research in recent years has indicated that they can be found in many body fluids. EVs have been identified and purified from blood, saliva, urine, and cerebrospinal fluid (CSF) [20,21,22,23]. There is no doubt that these EV contents can be used for diagnosis: (1) By merging them with clinicopathological factors, TNM staging, survival rate, and health/tumor definition can be easily classified [24]. (2) The properties of EVs have been used to control the rate and yield of exocytosis through gene transfection/infection, chemical compounds, growth factors, and genetic alteration events [25]. (3) EVs have targeting and homing abilities, can be modulated for their release area and target the tumor microenvironment or immune system, and have been regarded as a branch of biomedical imaging [26]. (4) Although EVs do not have the ability to replicate, they can be used as drug carriers for mass production and collection, creating more options for combination therapy [27].

The artificial manufacture of EVs has become a powerful technique to produce EVs on a large scale. Unlike traditional collection procedures, artificial methods can increase yield and reduce the scale of cell preparation and reuse. The methods of artificially promoting EV production are generally divided into physical and biochemical treatments. The physical methods can be used to produce EVs on a large scale through, for instance, an external electric field or ultrasonic vibration. The external electric field is used to stimulate the cells to release EVs. The application is based on the theory of electroporation and was developed as a new artificial manufacturing method called nano-electroporation (NEP) [28]. The NEP system can be applied to a monolayer of donor cells culturing on the surface of a chip, with the surface of the chip containing a series of nanochannels. The negative electrode is in contact with the bottom reservoir containing the cargo solution, which means that this method can also package the target cargo into EVs. After applying transient electrical pulses to cells, this NEP technique can produce large amounts of EVs. In addition, other studies have shown that cells treated with ultrasound stimulation can produce a larger quantity of EVs. The cells are repeatedly exposed to ultrasonic vibration for 10 min and then incubated for 30 min after excitation, and EV production can be expanded by 8–10 times. Through massive parallelization, Ambattu et al. used low-cost surface-reflected bulk waves to mass-produce EVs [29]. The biochemical treatments for EV production include changing the content of the cell culture medium, adding additional proteins that stimulate secretion of EVs, and genetic engineering of the gene structure of donor cells to secrete EVs on a large scale.

## 3. Modulation and Regulation Mechanisms for EVs and Their Components

### 3.1. ESCRT Complex

The formation of EVs involves a complicated mechanism and requires many steps. The core subunit is the endosomal sorting complex (ESCRT), which is composed of four main complexes (ESCRT-0, -I, -II, and -III) (Table 1). Several specific molecules are transported into the intraluminal region of the multivesicular bodies (MVBs) [30,31]. These subunits of ESCRT have unique roles in the exosome biogenesis. ESCRT-0 is a multivesicular reaction that can bind to the cell surface and accumulate receptors and/or ubiquitinated proteins. ESCRT-0 needs phosphoatidylinositol 3-phosphate to activate, directly interacts with the prosaposin domain, and then recruits the ESCRT-I subunit tumor susceptibility 101 (TSG101) to form the ESCRT-I complex [32]. In addition, ubiquitination assists ESCRT-I protein to activate and convert the bridge to connect between ESCRT-0 and ESCRT-II complexes. The ESCRT-II complex can deliver ubiquitinated proteins to endosomal membranes and plays a role in the biogenesis of multivesicular bodies. The targets are transferred from ESCRT-0 to ESCRT-I and then to ESCRT-II [33]. ESCRT-II and ESCRT-III form a cascade reaction in which the cargo containing vesicles is clamped. Therefore, ESCRT-III is responsible for cargo sorting, concentration, vesicle lysis, and material recycling [34]. ESCRT machinery also relies on several accessory proteins, including Vps4-Vta-1 complex and Bro1/ALIX proteins [34]. Once exocytosis is complete, the Vps4-Vta-1 protein strips other ESCRT components from the membrane, while the Bro1/ALIX protein helps to recruit deubiquitinases to the ESCRT-III [35]. This modification can remove the established ubiquitin tag and put an end to the exocytosis process.

### 3.2. RAB Family

Among the superfamily of Ras-like small GTPase, Rab GTPase is the largest family and has an essential function in the regulation of membrane identity and vesicle formation in the entire process of transportation in human physiology [51]. Dysregulation of Rab GTPases is associated with diverse inherited disorders. Among these disorders, neuron-associated diseases, including Parkinson’s disease and Huntington’s disease, are related to the imbalance of membrane trafficking regulated by Rab, which affects the neuron connections in the synapses [52]. Rabs can also regulate the translocation of Glut4 (a glucose transporter), which is associated with the pathogenesis of type 2 diabetes [53]. The key mediators necessary for innate immunity, such as the phagocytosis of intracellular pathogens, also require Rab members [54]. In cancer, the role of the Ras proto-oncogene in tumorigenesis has been widely discussed. However, how the Rab GTPases contribute to cancer progression remains largely unknown and is worth further investigation. To date, the most common endocytic-related Rabs are reported to be Rab5, Rab21, and Rab25 [55]. Among them, Rab25, which functions as a director in modulating integrin-recycling vesicle movement, has been well characterized. It is worth noting that Rab25 is also involved in the cell movement of epithelial cells, such as transformation and motility, which are correlated to tumor progression. The aggressiveness activity of female-related cancer types, such as breast and ovarian cancer, are all related to Rab25 overexpression. Interestingly, as a controversial factor, Rab25 serves as a tumor suppressor in colon cancer [56]. Another member, Rab5, was found to participate in the fusion of the early endosome process and regulate cell survival and migration by integrating with caspase 8, and the differential expression of Rab5 in cancers has been reported [57].

Only a few RABs are involved in exocytosis (RAB3/8/26/27); other family members participate in early endosomes (RAB4/5/10/17/21/23/35), late endosomes (RAB7/9), autophagosomes (RAB2/7/24/27/33), recycling endosomes (RAB11/13/17/25/35), and in anchoring to the ER–Golgi intermediated compartment (RAB2/18) [58] (Figure 1).

### 3.3. Genetic Alterations

Many tumor cells are prone to gene mutations throughout their development. Upon drug treatment, the selective pressure eventually sieves out cells with mutations showing drug-resistance capability. There are various genetic alterations reported in CRC. Some of these hotspots and events are highly correlated with the formation of EVs and exocytosis, thereby leading to increased malignancies and reducing the survival of patients. According to previous profiles, KRAS, PIK3CA, BRAF, EGFR, and ERBB2 are the most frequent gene alterations in CRC patients [59]. Isogenic colorectal cancer cell lines have been demonstrated to regulate EV cargo content (miRNAs, circRNAs, mRNAs, and proteins) in a KRAS-dependent manner [60,61,62,63]. Interestingly, RAB13 is not one of the RAB members that controls exosomes, but it has been proven to promote EV production in the KRAS mutation model [64]. PIK3CA mutant status is one of the hallmarks of CRC. Through next-generation sequencing analysis, approximately 30% of PIK3CA mutants were found in CRC patients [65]. PIK3CA affects the PI3K/Akt/mTOR pathway, and there is currently no effective targeting drug [66]. This pathway increases the secretion of EVs, and EVs containing PIK3CA can transfer from malignant cells to other cells for proliferation. There are also EVs that promote PIK3/Akt signaling and the exchange of substances (proteins and lipids) [67]. In addition, although there are targeted therapeutic compounds for EGFR (HER1) and ERBB2 (HER2) (cetuximab/panitumumab and trastuzumab/pertuzumab, respectively), they are not efficient in cases of mutations and continuously activate the downstream signals, including the RAS/RAF/MEK and PI3K/Akt/mTOR axes [68]. Moreover, in colon cancer, the V600E mutant of BRAF subsequently activates the RAS/RAF/MEK pathway, leading to a poor prognosis and strong exocytosis. Most importantly, the high CpG island methylator phenotype (CIMP-H) occurs simultaneously with MSI-H, KRAS_mut_, TP53_mut,_ BRAF_mut_, and various genetic alterations. The result is the overexpression and continuous production of exocytosis, which are beneficial to cancer progression.

## 4. Role of EVs in Colorectal Tumorigenesis

According to previous studies, the subunits of the ESCRT complexes and members of the RAB family that are involved in exocytosis have been identified (Table 1). These components have been investigated in colon cancer research. It has been reported that the expression levels of some candidates are varied during colon tumorigenesis. According to these reports, some genes are important for their prognostic value and can increase the hazard ratio of patients. Most roles of ESCRTs and RABs in cancer progression are found to be overexpressed to confer cancer cells with diverse functions related to malignancy. Environmental stress revealing enhanced proliferation and metabolic reprogramming has been found in cancer cells showing resistance to chemotherapy/radiotherapy, and overexpression of exocytotic and EV-related genes/proteins has been observed in auxiliary cancer cells. Additionally, these altered expression profiles also persist in the metastatic cancer cells. In contrast, Vps4-Vta1 and ALIX act as gatekeepers to terminate exocytosis and, therefore, their function is often suppressed in malignant colorectal tumors [45,46]. Most RABs show excessive expression, coordinating the production and transportation of EVs, but there are also some redundant trends (Table 1). Several RABs show inconsistent indications in colon cancer studies [49,50] and need to be studied and classified in detail in patient cohorts.

The large amount of biological information carried by EV particles has been shown to be related to tumorigenesis and endows cancer cells with different phenotypes (Table 2). The most common ones contain many non-coding RNAs and proteins. It has been confirmed that abundant miRNAs are related to colon cancer development, most of which are highly expressed to regulate a variety of cancer functions (chemoresistance, radioresistance, immune response, metastasis, and proliferation). In contrast, some tumor suppressor miRNAs have also been discovered [27,69,70,71]. lncRNAs and circRNAs directly control the expression and processing of miRNAs and affect tumor progression. As a well-known long non-coding RNA, lncCRNDE regulates multiple signaling pathways and functions [72]. It is considered as a diagnostic/prognostic predictor of colon cancer. The direct targets of these lncRNAs have been explored to elucidate the pathogenic mechanisms. For example, Liu et al. discovered that lncCRNDE has a positive correlation with IRX5 RNA [73]. Cheng and his group also reported that lncRNA GAS5 can inhibit colorectal cancer cell proliferation via the miR-182-5p/FOXO3a axis [74]. Similarly, both CCAT2-miR-145 and CCAL-β-catenin regulation are presented in colon cancer proliferation and progression [75,76]. Moreover, circHIPK3 interferes with miR-7 to promote the growth and metastasis of colorectal cancer and acts as an alternative to regulate miR-1207-5p/FMNL2 signaling [77,78]. Yang et al. claimed that circ-133 acts on the miR-133a/GEF-H1/RhoA axis to promote colon cancer metastasis [79]. Hon et al. stated that circ_0000338 plays a dual regulatory role in chemoresistance CRC [80]. Consistent with this, circRNAs also reflect aberrant exocytosis caused by genetic changes and participate in its regulation [62]. Some exosomal proteins can be used for blood screening, EV collection, and analysis of patient clinical indicators [81,82,83,84,85,86,87,88,89,90,91,92,93]. These proteins can also regulate single or multiple aspects of colon cancer. Other related proteins have been observed in these studies but are not involved in exocytosis, and thus we exclude them from discussion in this article (Figure 1).

On the other hand, compared with the exosome derived from cancer cells, exosomes derived from the immune system have different contributions. They can meditate crosstalk between immunity and regulate cancer progression. First, B cell-derived exosomes were identified, and then it was found that lymphocytes, dendritic cells (DCs), natural killer cells, mast cells, macrophages, and thymocytes can all produce exosomes [123]. Yan et al. has demonstrated that the exosomes of various immune cells contain their origins, target cells, consequences, and involved molecules [124]. According to these previous studies from the literature, exosomes derived from immune cells can effectively regulate the immune response (innate/adaptive) and provide cytotoxic effects against cancer cells. Moreover, EVs derived from immune cells are known to exert similar functions as those of their parent immune cells, and the development of activated immune-derived EVs can also be used for immunotherapy applications [125]. Conversely, if EVs are derived from immune cells that are known to promote tumor development (such as TAMs), they can instead increase the malignancy of cancer phenotype in many aspects [126,127].

## 5. Available Omics Datasets of EVs for CRC

Previous studies have stated that EVs package a variety of fragments containing biological information, including DNA, RNA, and proteins [128]. These substances rearrange the tumor microenvironment and regulate the immune system and metabolic events, thereby promoting tumor colonization, proliferation, and metastasis [129]. The extraction of EVs, determination of their contents, and classification could be utilized to evaluate whether they can be applied as clinical parameters for prognosis and to investigate the underlying molecular mechanisms.

In the multi-omics category, many profiles of EVs and exosomes have been established (Table 3). These datasets provide potential candidates through hierarchical analysis and statistical calculations, which can accurately determine the corresponding clinical events of colorectal cancer. The core data or clustering that meets the cut-off/fold change can also predict potential upstream/downstream pathways, as well as the transcription factors involved and the consequences of gene/drug regulation. Ohshima et al. observed and verified the quality of exosomes through transmission electron microscopy (TEM) and Western blotting (CD29, Tsg101, Alip1, and Bip) [130]. Compared with the cultured cell lines and their media, they determined that in the exosomes several members of the let-7s family of miRNAs undergo significant changes. Although they illustrated the proposed model with the metastatic gastric cancer cell AZ-P7a, they also established profiles in colorectal cancer cells (SW480 and SW620) [130]. In the study by Otaga-Kawata et al., serum was collected to extract exosome-enriched fractions from CRC clinical patients. In addition to let-7, they found six more candidates (miR-1229, miR-1246, miR-150, miR-21, miR-223, and miR-23a) that were upregulated compared to healthy controls [94]. These candidates have been compared with current biomarkers of colorectal cancer (e.g., carbohydrate antigen (CA19-9) and carcinoembryonic antigen (CEA)). Their results show that these candidates are of good specificity and sensitivity for use as additional biomarkers for CRC.

Genetic alterations have always been considered an important issue in tumorigenesis and affect the secretion and exocytosis of EVs. KRAS is a typical mutation event in both CRC cell lines and clinical specimens. Cha and her team revealed that small RNA composition is correlated with KRAS status by using the wild-type KRAS cells DKs-8, mutant KRAS cells DKO-1, and their counterpart cells DLD-1 as models [60]. They listed the increase or decrease in gene expression in a single cell model and showed the common signatures in their study. The Dou and Hinger groups also chose similar models for comparison [62,63]. They detected some circular RNAs and long non-coding RNAs. These results are also consistent with the proposed model according to which EVs and exosomes contain various biomessenger fragments. This evidence also suggests that EVs are suitable for development as prediction markers. Furthermore, Yoshii et al. explored the differences in exosomal functions and miRNA expression levels in the absence of TP53 (GSE120013).

Other EV studies in CRC used various colorectal cancer cells as models. Chiba recruited exosomes from SW480 cells (GSE68979), and Ji et al. distinguished three subtypes of EVs from the human LIM1863 cancer cell line [131]. Moreover, Sun and his group reported their results in HT-29 and SW-948 cells [134,135]. Some stem cell markers (CD44v6, Tspan8, CD151, and Claudin7) have been manipulated in exosome research. In the CRC cell models, Tubita et al. also created a series of multi-omics datasets and mentioned that several miRNAs (miR-6127, miR-6746-5p, and miR-6787-5p) can be regulated in the pre-metastatic niche of colon cancer [136]. RNA-binding proteins and other cellular proteins have also been observed, and their changes have been confirmed in exosomes [137]. The EVs’ core omics profiles are not limited to the collected species and sources; transcriptomics and RNA-seq data can also be completed with patient specimens, xenografted tissues, and animal models/cell lines. These libraries include comparisons of before and after treatments, single target modulation, and normal distribution between clinical cohorts [132,133]. In addition, the multiple omics established in the Mus musculus also provide a better biological reservation and efficacy evaluation perspective for exosomal research (GSE101950, GSE101951, and GSE173202).

## 6. Current Combinations and Clinical Trials of EV-Based Carriers

Trials and combination therapies related to EVs are ongoing. The main reason is that, as we have mentioned, EVs have the characteristics of cargo loading and delivery. Through announced clinical trials (http://clinicaltrials.gov, accessed on 30 June 2021), various EV applications and combination therapies are being considered for colon cancer research (Table 4). Most of these clinical EV studies involve the use of patients’ blood samples for biomarker evaluation. These studies divide patients by healthy and tumor samples, primary/metastasis status, and other cancer subtypes to investigate the differences. In addition, they further identified components with the available parameters, including macromolecules, integrins, and metalloproteases. Furthermore, several trials have begun to use EVs as drug carriers. These drugs include curcumin (NCT01294072), Toripalimab (NCT03927898), and AL3810 (NCT03260179). In addition, in 1989, Sidney Altman and Thomas Cech discovered that small RNA and DNA have functions such as ligand binding and activation and gene regulation. They could screen for secondary structure RNA or single-stranded DNA that would specifically bind to the target protein. These specialized small molecule nucleic acids are called aptamers. Aptamers can also be coupled to EVs to promote specific targeting, such as the dart method for EVs that compete with CD63-specific aptamers [138]. This modification can direct EVs to their targeting cells or tissues and deliver the drugs only to specific tissue or cells. Moreover, recent research has made progress in the development of nucleic acid drugs (such as mRNA). Several nucleic acid drugs have been approved for marketing. The clinical data of these potential blockbuster drugs have recently been released. There are also endless mergers, acquisitions, and product introduction transactions in the field of nucleic acid drugs. Due to the recent COVID pandemic, the development of mRNA vaccines has also received more attention. These nucleic acid drugs can also be embedded in EVs and delivered to target cells or tissues. Using EVs as carriers can provide the following improvements for the delivery of nucleic acid drugs: First, EVs can avoid rapid elimination of nucleic acid drugs and extend their half-life. Second, EVs can target specific sites or home in on primary colorectal tumors through small peptides, aptamers, and surface antibodies. Finally, EVs can reduce the drug escape efficiency of endosomes to make nucleic acid drugs more likely to stay in target cells [139,140,141] (Figure 2).

At present, data on the clinical use of EVs in cancer treatment are quite scarce. In addition to various patents, there are currently many restrictions, including purification, formulation, dosage, delivery, biodistribution, etc. Xu et al. have summarized the purification methods used in EVs. Except for synthetic polymer-based precipitation approaches that can achieve higher EV yields, most other methods can only get low to medium yields and quantities (μL–mL) [142]. In addition, many purification methods must deal with particles with sizes more than 70 nm, and this relatively large particle size will seriously affect its production. In a study on colon cancer, Dai et al. tried to collect ascites-derived exosomes (Aex) from 800 mL ascites and then, after combining them with granulocyte-macrophage colony-stimulating factor (GM-CSF), compared them with the control (Aex alone, 100~500 μg) [143]. They claimed that this strategy is feasible and safe and that future research should refer to and use it as a template for improvement. Moreover, the yield and composition of EVs also have a lot to do with the source. As Allelein et al. confirmed, heterogenous populations vary greatly. This phenomenon must exist in clinical samples [144]. In order to overcome these bottlenecks, we suggest that more studies need to sort out detailed information, including source cell type, extraction method, purity, modified vesicle type, etc. [145].

## 7. Future Prospects

In view of their role in known mechanisms of colon cancer, EVs can be used as therapeutic targets by inhibiting the formation, release, and uptake of EVs or by targeting a biologically active substance to reverse colorectal tumor phenotypes. EVs can also be applied as therapeutic agents. Unlike common drug delivery vehicles such as liposomes or nanoparticles, EVs have minimal immunogenicity and toxicity. In addition, they are produced by endogenous cells, have fewer rejection reactions, and can carry a variety of combinations (small molecule compounds, modified peptides, monoclonal antibodies, and antagomiRs). These advantages enrich the application of EVs as a breakthrough point in combination therapy.

EVs and cellular exocytosis are still difficult to modulate, and the complete mechanism remains to be investigated. Although many candidates are related to exosomal particles, omics profiles cannot reveal the degree of consistency/dependence between them. Which is the most critical role/event in exosomes has yet to be verified. At present, most EVs are limited to clinical prognosis and diagnosis. In future basic research on EVs, it is necessary to determine the exact target genes and downstream axis regulated by miRNAs. At the same time, the identified circRNA/lncRNA should also help to elucidate the relationship with miRNA. Finally, EV-based strategies also require a comprehensive biomedical imaging platform. Through real-time feedback to patients with colorectal cancer, the potential therapeutic value can be monitored and evaluated.

## Figures and Tables

**Figure 1 biomedicines-09-00931-f001:**
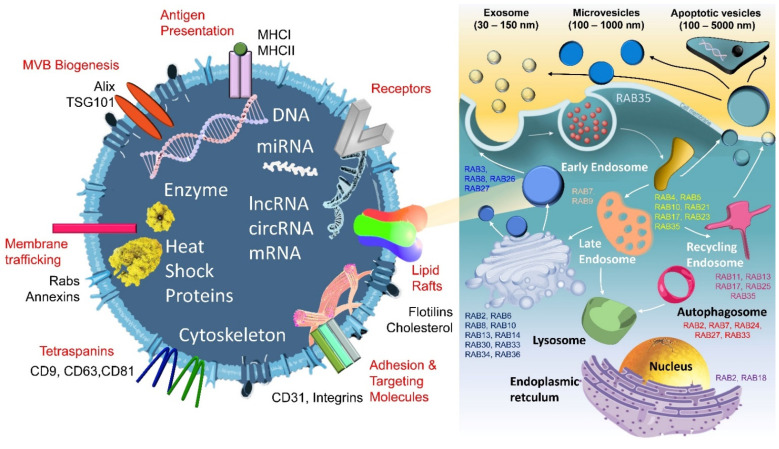
The classification of extracellular vesicles (EVs), the process of exocytosis, and components of EVs. Members of the RAB family regulate different EVs (early/late endosome, recycling endosome, autophagosome, etc.). EVs embed various substances and anchor receptors and factors on the membrane.

**Figure 2 biomedicines-09-00931-f002:**
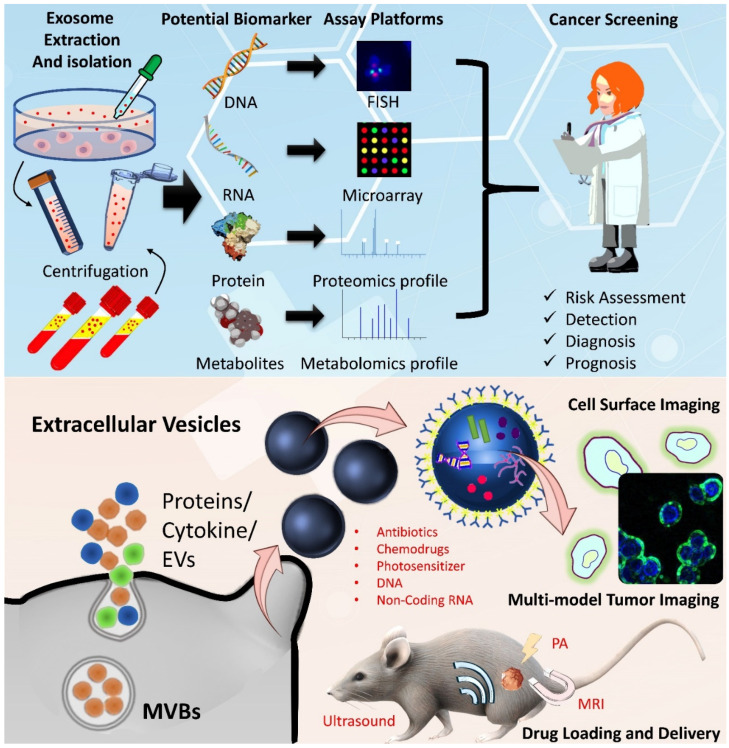
Current exosomal materials and application aspects. DNA, RNA, protein, and metabolites can be extracted from EVs in cancer cells, and potential clinical events (risk assessment, detection, diagnosis, and prognosis) can be analyzed. In animal models, EVs can be used to carry drugs as a strategy for biomedical imaging and therapy.

**Table 1 biomedicines-09-00931-t001:** Components of ESCRT complexes and regulators in colorectal cancer exocytosis.

Complex	Gene Symbol	Hazard Ratio	Expression	Ref.
ESCRT-0	VPS27 (HRS/HGS)	Uni: 2.27 Multi: 3.34	Up	[36]
	STAM1/2	----	Up	[37]
ESCRT-I	VPS23 (TSG101)	----	Up	[38]
	VPS28	----	----	-----
	VPS37A/B/C/D	----	Down	[39]
	MVB12A/B	----	----	----
	UBAP1	----	Down	[40]
ESCRT-II	VPS22 (SNF8, EAP30)	----	----	-----
	VPS25 (EAP20)	----	----	-----
	VPS36 (EAP45)	----	----	-----
ESCRT-III	VPS2A/B (CHMP2A/B)	----	Up	[41]
	VPS20 (CHMP6)	----	----	-----
	VPS24 (CHMP3)	----	----	-----
	SNF7A/B/C (CHMP4A/B/C)	----	Down	[42]
	VPS60 (CHMP5)	----	Up	[43]
	DID2A/B (CHMP7, CHMP1A/B)	----	Down	[42]
	IST1 (OLC1)	Uni: 10.43 Multi: 7.9	Up	[44]
Vps4-Vta1	VPS4A/B (SKD1)	----	Down	[45]
	VTA1 (LIP5)	----	----	----
Bro1/ALIX	ALIX (PDCD6IP)	----	Down	[46]
RABs	RAB3	Uni: 2.58 Multi: 2.39	Up	[47]
	RAB8	----	Up	[48]
	RAB26	----	Up	[47]
	RAB27	Multi: 0.45	Up Down	[49] [50]

**Table 2 biomedicines-09-00931-t002:** Vesicle-related substances carried by colorectal cancer.

Category	Name	Express	Function	Ref.
miRNA	miR-1246	Up	Diagnosis	[94]
	miR-125a-3p	Up	Diagnosis	[95]
	miR-150-5p	Down	Diagnosis	[71]
	miR-17-5p/miR-92a-3p	Up	Metastasis	[96]
	miR-181a-5p	Down	Metastasis	[97]
	miR-19a	Up	Recurrence	[98]
	miR-193a	Up	Metastasis	[99]
	miR-196b-5p	Up	Chemoresistance	[100]
	miR-203	Up	Metastasis	[101]
	miR-210	Up	Chemoresistance/metastasis	[102]
	miR-21	Up	Recurrence/chemoresistance/metastasis	[94]
	miR-21-5p/miR-1246/miR-96-5p/miR-1229-5p	Up	Chemoresistance	[103]
	miR-23a	Up	Diagnosis	[94,104]
	miR-25-3p	Up	Metastasis	[105]
	miR-30d-5p	Up	Metastasis	[97]
	miR-301a	Up	Diagnosis	[104]
	miR-486-5p	Up	Diagnosis	[106]
	miR-548c-5p	Down	Metastasis	[69]
	miR-638	Down	Metastasis	[107]
	miR-6803-5p	Up	Diagnosis	[108]
	miR-6869-5p	Down	Metastasis	[70]
	miR-92a-3p	Up	Chemoresistance	[109]
	Let-7b-3p/miR-139-3p/miR-145-3p	Up	Diagnosis	[110]
circRNA	circHIPK3	Up	Diagnosis/metastasis	[78,111]
	circ-133	Up	Metastasis	[79]
	ciRS-122	Up	Chemoresistance	[112]
	hsa-circ_0004771	Up	Diagnosis	[113]
	hsa-circ_0000338	Up	Chemoresistance	[80]
	circRTN4	Up	Chemoresistance	[114]
lncRNA	lncRNA CRNDE-h	Up	Recurrence/chemoresistance/ metastasis/diagnosis	[72,115]
	lncRNA GAS5	Up	Diagnosis	[116]
	LNCV6_116109	UP	Diagnosis	[117]
	LNCV6_98390	Up	Diagnosis	[117]
	LNCV6_84003	UP	Diagnosis	[117]
	LNCV6_98602	Up	Diagnosis	[117]
	LNCV_108266	Up	Diagnosis	[117]
	LNCV6_38772	Up	Diagnosis	[117]
	lncRNA 91H	Up	Recurrence	[118]
	lncRNA CCAT2	Up	Diagnosis	[119]
	lncRNA H19	Up	Chemoresistance	[120]
	lncRNA CCAL	Up	Chemoresistance	[75]
	lncRNA PVT1	Up	Metastasis	[121,122]
Protein	Hsp60	Up	Diagnosis/proliferation	[81]
	GPC1	Up	Diagnosis/metastasis	[84]
	CD147	Up	Diagnosis	[88]
	CPNE3	Up	Diagnosis	[86]
	TAG72	Up	Chemoresistance	[91]
	S100A9	Up	Recurrence	[89]
	SPARC	Up	Diagnosis/angiogenesis	[93]
	LRG1	Up	Diagnosis	[93]
	CEA	Up	Diagnosis/metastasis	[92]
	IRF-2	Up	Metastasis	[87]
	Wnt	Up	Chemoresistance	[83]
	CAPS1	Up	Metastasis	[90]
	MAGEA3	Up	Diagnosis/metastasis	[82,85]

**Table 3 biomedicines-09-00931-t003:** Available microarray chips and RNA-seq datasets of exosomal research in colorectal cancer from the GEO website.

	Array Chip and RNA-Seq Platform	Species	Objects	Ref.
GSE21350	Agilent-021827. Human miRNA Microarray G4470C	Human	Cell line (SW480,SW620)	[130]
GSE39833	Agilent-021827. Human miRNA Microarray G4470C	Human	CRC patients’ serum	[94]
GSE40246	Agilent-021827. Human miRNA Microarray G4470C	Human	CRC patients’ serum	[94]
GSE67004	Illumina HiSeq 2000	Human	Cell line (DKO-1,DLD-1,DKs-8)	[60]
GSE68979	Agilent-019052 Homo sapiens 45K	Human	Cell line(SW480)	N/A
GSE72577	Illumina HiSeq 2000	Human	Cell line (DKO-1,DLD-1,DKs-8)	[62]
GSE87839	Applied Biosystems Taqman Array Human Micro A+B Cards Set v3.0	Human	Cell line (LIM1863)	[131]
GSE100063	Illumina HiSeq 2000	Human	CRC patients’ blood	[132]
GSE101950 GSE101951	Illumina HiSeq 2500	Mus	Cell line (CT26)	N/A
GSE114316 GSE114317 GSE114318	3D-Gene Human Oligo Chip 25K V2.13D-Gene Human miRNA V21_1.0.0	Human	Colorectal xenografts	[133]
GSE115114	Illumina NextSeq 500	Human	CRC patients	N/A
GSE116589	Illumina HiSeq 3000	Human	CRC patients	N/A
GSE119031	Agilent-040150 EMBL-rel18_30rep 031181	Human	Cell line (HT-29,SW-948)	[134,135]
GSE119032 GSE119033	Agilent-070156 Human_miRNA_ V21.0_Microarray 046064	Human	Cell line (HT-29,SW-948)	[134,135]
GSE120013	3D-Gene Human miRNA V21_1.0.0	Human	Cell line (HCT116)	N/A
GSE121964	Illumina HiSeq 2000	Human	Cell line (DKO-1,DLD-1,DKs-8)	[63]
GSE123708 GSE123709 GSE123710	Affymetrix Multispecies miRNA-4 ArrayAffymetrix Clariom S Assay HT, Human	Human	Cell line (HCT116,SW480)	[136]
GSE125905	Illumina HiSeq 2000	Human	Cell line (DKO-1,Gli36)	[137]
GSE173202	Illumina HiSeq 2000	Mus	Cell line (CT26,MC38)	N/A

**Table 4 biomedicines-09-00931-t004:** Current clinical trials involve EVs for colorectal cancer.

Number	Participants	Phase	Status	Application
NCT01294072	35	Phase I	Recruiting	Treatment
NCT03432806	80	----	Recruiting	Diagnostic
NCT04394572	75	----	Recruiting	Diagnostic
NCT04523389	172	----	Recruiting	Diagnostic
NCT04298398	108	----	Not yet recruiting	Diagnostic
NCT04394572	75	----	Recruiting	Diagnostic
NCT04523389	172	----	Recruiting	Diagnostic
NCT03927898	40	Phase II	Recruiting	Treatment
NCT02439008	28	----	Terminated	Diagnostic
NCT03260179	60	Phase I	Unknown	Treatment

## Data Availability

Not applicable.

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
