# Peer review of "Exosomal Components and Modulators in Colorectal Cancer: Novel Diagnosis and Prognosis Biomarkers"

_biomedicines, 2021, doi:10.3390/biomedicines9080931_

Round 1

Reviewer 1 Report

This review by Yu-Chan Chang and collaborators summarizes the function and application of Extracellular vesicles (EVs) in the colon cancer research field. They provide an update regarding the potential values for clinical applications of EVs. Moreover, they illustrate the specific markers, mediators, and genetic alterations of EVs in CRC. Furthermore, they outline the vital markers presented in the EVs and discuss their plausible uses in colon cancer patient therapy in combination with the currently used clinical strategies.

Comments/questions:

The topic is of interest. The authors provided an update regarding the importance of EVs in Colon cancer and discussed their potential uses in CRC patient therapy. The manuscript is well written. 

What is known about the EVs produced by immune cells associated to colon cancer? In terms of composition, is there any difference between EVs from macrophages and vesicles produced by tumor cells? Is there any correlation between EVs composition (biomarkers expression) and colon cancer progression? Please, if it is possible provide some more information.

Author Response

Reviewer 1:

This review by Yu-Chan Chang and collaborators summarizes the function and application of Extracellular vesicles (EVs) in the colon cancer research field. They provide an update regarding the potential values for clinical applications of EVs. Moreover, they illustrate the specific markers, mediators, and genetic alterations of EVs in CRC. Furthermore, they outline the vital markers presented in the EVs and discuss their plausible uses in colon cancer patient therapy in combination with the currently used clinical strategies.

Comments/questions:

The topic is of interest. The authors provided an update regarding the importance of EVs in Colon cancer and discussed their potential uses in CRC patient therapy. The manuscript is well written. 

Answer: We thank reviewer #1 for the encouragement and suggestions.

What is known about the EVs produced by immune cells associated to colon cancer? In terms of composition, is there any difference between EVs from macrophages and vesicles produced by tumor cells? Is there any correlation between EVs composition (biomarkers expression) and colon cancer progression? Please, if it is possible provide some more information.

Answer: We have added a description of EBs derived from the immune system (including macrophages) and discussed the different characteristics of cancer derived EVs. “On the other hand, compared with the exosome derived from cancer cells, exosomes derived from the immune system have different contributions. They can meditate crosstalk between immunity and regulate cancer progression. First, B cell-derived exosomes have been identified, and then it was found that lymphocytes, dendritic cells (DCs), natural killer cells, mast cells, macrophages and thymocytes can all produce exosomes [123]. Yan et al. has illustrated that the exosomes of various immune cells contain their origins, target cells, consequences and involved molecules [124]. From those works of literature, exosomes derived from immune cells can effectively regulate the immune response (Innate/adaptive) and provide cytotoxic effects against cancer cells. Moreover, EVs derived from immune cells are known to exert similar functions as that of their parent immune cells, and the development of activated immune-derived EVs can also be used for immunotherapy applications [125]. Conversely, if EVs are derived from immune cells that are known to promote tumor development (such as TAMs), it can instead increase the malignancy of cancer phenotype in many aspects [126,127].

Reviewer 2 Report

In the manuscript “Exosomal Components and Modulators in Colorectal Cancer: Novel Diagnosis and Prognosis Biomarkers” Chen and co-authors review current knowledge on function, regulation, genetic alterations of EVs in colorectal cancer; moreover, they discuss potential theranostic clinical applications of EVs.

General comment:

The review (in particular the tables) is informative.

Specific comments:

  • The titles of some sections should be modified to reflect more accurately section content and improve manuscript organization.
  • Figures need to be revised.
  • Current limitations to EVs clinical translation for colon cancer (including issue related to isolation of suitable amounts of EV for clinical use, clinical dose determination, route of delivery and biodistribution) should be briefly addressed.

Minor issues:

  • Page 2 Line 15: abbreviations are used in the text should be defined in the text at first use (for instance epithelial–mesenchymal transition (EMT) line 15; 5-FU line 20)
  • Page 2 Line 16: “The EMT process is accompanied by drug resistance, cell morphology, metastasis..”. Cell morphology changes.
  • Page 2 Line 46: The section entitled “EVs application in CRC”. This section address some general information of EV, but it is not focused on CRC as one might expect from the title.
  • Page 3 Line 25 : Please, define the what you consider “Implementation and supervision mechanism of EVs”
  • Page 6 Line 44. In my opinion also the title “Available EV-related small molecule compounds”, should be clarified and revised to reflect more accurately the description of the section.
  • Page 7 Line 32. “Although there are many candidates related to exosomal particles, most of them are macroscopic changes exposed by high-throughput and omics profiles, and the specific microscopic effects are still unclear”. Please, rephrase.
  • Figure 1: does not illustrate the classification of extracellular vesicles
  • Figure 2 top panel: it is too generic and it is not clear whether it refers specifically to EV
  • Figure 2: bottom panel: is not clear the reference to “proteins”. Does the image of tumor imaging refer to cells or EVs? Not clear that the drug loading is applied to isolated EVs

Author Response

Reviewer 2:

In the manuscript “Exosomal Components and Modulators in Colorectal Cancer: Novel Diagnosis and Prognosis Biomarkers” Chen and co-authors review current knowledge on function, regulation, genetic alterations of EVs in colorectal cancer; moreover, they discuss potential theranostic clinical applications of EVs.

General comment:

The review (in particular the tables) is informative.

Answer: We thank the reviewer #2 for the encouragement and suggestions.

Specific comments:

  • The titles of some sections should be modified to reflect more accurately section content and improve manuscript organization.

Answer: We revised some subtitles and adjusted the order of paragraphs to improve this manuscript.

  • Figures need to be revised.

Answer: In response to several weaknesses, we modified figures 1-2. Thank you.

  • Current limitations to EVs clinical translation for colon cancer (including issue related to isolation of suitable amounts of EV for clinical use, clinical dose determination, route of delivery and biodistribution) should be briefly addressed.

Answer: At present, data on the clinical use of EVs in cancer treatment is quite scarce. In addition to various patents, there are currently many restrictions, including purification, formulation, dosage, delivery, and biodistribution, etc. Xu et al have summarized the purification methods used in EVs. Except for synthetic polymer-based precipitation approaches that can achieve higher EV yields, most other methods can only get low to medium yields and quantities (μl-ml) [142]. In addition, many purification methods must deal with particles with size more than 70nm, and this relative large particle size will seriously affect its production. In colon cancer, Dai et al. tried to collect ascites-derived exosomes (Aex) from 800 ml ascites, and then combined with granulocyte-macrophage colony stimulating factor (GM-CSF), compared with control (Aex-alone, 100~500μg)[143]. They claimed that this strategy is feasible and safe, the future research should refer to and use this as a template for improvement. Moreover, the yield and composition of EVs also have a lot to do with the source. As Allelein et al confirmed, heterogenous populations vary greatly. This phenomenon must exist in clinical samples [144]. In order to solve these bottlenecks, we suggest that most studies need to sort out detailed information, including source cell type, application route, isolation, modified vesicle type, etc.[145].

Minor issues:

  • Page 2 Line 15: abbreviations are used in the text should be defined in the text at first use (for instance epithelial–mesenchymal transition (EMT) line 15; 5-FU line 20)

Answer: We thank reviewer #2 for the kind reminder. We added their full names and abbreviations when we first used EMT and 5-Fu in this manuscript.

  • Page 2 Line 16: “The EMT process is accompanied by drug resistance, cell morphology, metastasis..”. Cell morphology changes.

Answer: We thank reviewer #2 for the suggestion. We have modified it.

  • Page 2 Line 46: The section entitled “EVs application in CRC”. This section address some general information of EV, but it is not focused on CRC as one might expect from the title.

Answer: We thank reviewer #2 for the critical comments. We revised the subtitle to “General definition, classification, and application of EVs”.

  • Page 3 Line 25 : Please, define the what you consider “Implementation and supervision mechanism of EVs”

Answer: We thank reviewer #2 for the critical comments. We revised the subtitle to “Modulation and regulation mechanism for EVs and its components”.

  • Page 6 Line 44. In my opinion also the title “Available EV-related small molecule compounds”, should be clarified and revised to reflect more accurately the description of the section.

Answer: Thank you reviewer #2 for your criticism. We revised the subtitle to “Current combinations and clinical trials of EV-based carriers”.

  • Page 7 Line 32. “Although there are many candidates related to exosomal particles, most of them are macroscopic changes exposed by high-throughput and omics profiles, and the specific microscopic effects are still unclear”. Please, rephrase.

Answer: We rephrased the sentence as “Although there are many candidates related to exosomal particles, omics profiles cannot reveal the degree of consistency/dependence between them. Which one is the most important role/event in exosomes has yet to be verified.”

  • Figure 1: does not illustrate the classification of extracellular vesicles

Answer: We have modified the figure 1 and added the classification of EVs in the upper right corner of the figure.

  • Figure 2 top panel: it is too generic and it is not clear whether it refers specifically to EV
  • Figure 2: bottom panel: is not clear the reference to “proteins”. Does the image of tumor imaging refer to cells or EVs? Not clear that the drug loading is applied to isolated EVs

Answer: We modified the figure 2 and added icons for exosomes extraction and isolation to illustrate that these analysis approaches focus on EVs. In the bottom panel, we have also made appropriate adjustments to describe that using EV as a carrier, cells and animals produce biological images after ingesting EV.
